# Tumor Microenvironment Modulation by Cancer-Derived Extracellular Vesicles

**DOI:** 10.3390/cells13080682

**Published:** 2024-04-15

**Authors:** Artem Ten, Vadim Kumeiko, Vladislav Farniev, Huile Gao, Maxim Shevtsov

**Affiliations:** 1School of Medicine and Life Sciences, Far Eastern Federal University, 690922 Vladivostok, Russia; ten.arteym@gmail.com (A.T.); frear.far@mail.ru (V.K.); vkumeiko@yandex.ru (V.F.); 2Key Laboratory of Drug-Targeting and Drug Delivery System of the Education Ministry, Sichuan Engineering Laboratory for Plant-Sourced Drug and Sichuan Research Center for Drug Precision Industrial Technology, West China School of Pharmacy, Sichuan University, Chengdu 610064, China; gaohuile@scu.edu.cn; 3Laboratory of Biomedical Nanotechnologies, Institute of Cytology of the Russian Academy of Sciences, Tikhoretsky Ave., 4, 194064 St. Petersburg, Russia; 4Personalized Medicine Centre, Almazov National Medical Research Centre, Akkuratova Str., 2, 197341 St. Petersburg, Russia; 5Department of Radiation Oncology, Technishe Universität München (TUM), Klinikum Rechts der Isar, Ismaninger Str., 22, 81675 Munich, Germany

**Keywords:** tumor microenvironment, exosome, cancer stem cell, cancer-associated fibroblast, tumor-associated macrophage, cancer-associated neutrophil, cancer-associated endothelial cells

## Abstract

The tumor microenvironment (TME) plays an important role in the process of tumorigenesis, regulating the growth, metabolism, proliferation, and invasion of cancer cells, as well as contributing to tumor resistance to the conventional chemoradiotherapies. Several types of cells with relatively stable phenotypes have been identified within the TME, including cancer-associated fibroblasts (CAFs), tumor-associated macrophages (TAMs), neutrophils, and natural killer (NK) cells, which have been shown to modulate cancer cell proliferation, metastasis, and interaction with the immune system, thus promoting tumor heterogeneity. Growing evidence suggests that tumor-cell-derived extracellular vesicles (EVs), via the transfer of various molecules (e.g., RNA, proteins, peptides, and lipids), play a pivotal role in the transformation of normal cells in the TME into their tumor-associated protumorigenic counterparts. This review article focuses on the functions of EVs in the modulation of the TME with a view to how exosomes contribute to the transformation of normal cells, as well as their importance for cancer diagnosis and therapy.

## 1. Introduction

Extracellular vesicles (EVs) are lipid-membrane-bound secretory structures that play a crucial role in cancer biology, mediating intercellular communication between tumor and normal cells within the tumor microenvironment (TME) [1]. According to the International Society for Extracellular Vesicles (and MISEV2024 update), EVs are classified according to various criteria, as follows: (a) biogenesis pathway—endosome-origin “exosomes” or plasma-membrane-derived “ectosomes” (i.e., microparticles/microvesicles); (b) size—small (<100 nm) or large (>200 nm); (c) biochemical composition (CD63+/CD81+-EVs, annexin-A5-stained EVs, etc.); and (d) according to the cell of origin or physiological processes (podocyte EVs, hypoxic EVs, large oncosomes, apoptotic bodies, etc.) [2]. The most common classification is related to the pathway of EV biogenesis: exomeres, exosomes, ectosomes (i.e., microvesicles), migrasomes, apoptotic bodies, and oncosomes (Figure 1) [3]. The constitutive contents of exosomes include various RNAs (e.g., miRNA, lncRNA, and circRNA), DNA fragments, transcription factors, signaling proteins, and many other molecules [4]. EVs have a number of advantages as mediators. Firstly, because of their lipid membrane bilayer, they preserve the contents from external influences and allow them to be activated only inside the recipient cell. Secondly, surface markers enable the targeted delivery of the exosomal content [5]. The uptake of extracellular vesicles by recipient cells is mediated by several mechanisms including phagocytosis, clathrin- and caveolin-mediated endocytosis, macropinocytosis, and membrane fusion [6].

Tumor cells under stressful conditions are characterized by an increased level of exosome release [7]. Within a tumor, exosomes are involved in modulating the tumor microenvironment. Presumably, tumor-derived EVs ensure the heterogeneity of normal cell populations through the delivery of physiologically active cargo. The heterogeneity of cancer cells has long been documented [8]. However, in addition to tumor heterogeneity, there is also heterogeneity in other cell groups [9]. The cohabitation of different types of cells within the TME under harsh conditions creates several types of interactions. Merlo et al. identified five types of interactions: competition, predation, parasitism, mutualism, and commensalism [10]. Transformation of normal cells in this case can be considered a means of creating the most favorable conditions for the tumor and, accordingly, changing the interaction toward a symbiotic relationship. Considering that stable populations of transformed normal cells, such as cancer-associated fibroblasts (CAFs), tumor-associated macrophages (TAMs) [11], tumor-associated neutrophils (TANs), and others, are found within the solid tumor, one could assume that during tumor-driven selection, they have some advantage over others (Table 1).

However, there are no data on further modulation of already-transformed normal cells. Thus, fibroblasts transformed into CAFs have no well-defined phenotype that can be used to describe this cellular subpopulation. This tumor cell–CAF interaction is the only documented example of mutualism in a tumor beneficial for both cell types [33]. Accordingly, it has been shown that exosomal LncRNA LINC00659 of CAF enhances colorectal cancer cell progression via the miR-342-3p/ANXA2 axis [34]. On the other hand, the heterogeneous TME is influenced by the antitumor responses of the organism, mainly associated with the immune system [35]. Thus, the formation of cellular subpopulations of transformed normal cells within the TME occurs under the influence of various factors derived from the tumor itself and the organism (Figure 2). The question of the direction of these factors remains unclear. Additionally, it is worth considering that the release of EVs by tumor cells also depends on external conditions. Thus, hypoxic conditions increase the release of EVs by tumor cells which results in increased angiogenesis and migration [36]. The molecular mechanisms of the influence of EV cargo on the heterogeneity of specific cell types are discussed in the following sections.

In addition to the exosomal cross-talk mechanism, other mechanisms of cancer-associated cell transformation have been described, including close intercellular contacts, paracrine signaling pathways, growth factors (BMP2 and BMP4), and stress-related conditions (hypoxia, low pH, etc.). Kyuno et al. described a significant impact of the cell–cell junction on the epithelial-to-mesenchymal transition of cancer cells [37]. The main molecular player among the soluble factors is considered to be TGF-β [38]. It is secreted by different cells of the TME and induces significant modulation of the tumor environment, bringing about a cascade of changes to tumorigenesis [39]. Although various stress-related conditions have been identified, including hypoxia, oxidative stress, and other factors, it is still quite difficult to estimate the input of each factor for TME transformation, as these factors are usually simultaneously present in the tumor and have a synergistic effect on the infiltrating cells [40,41,42,43].

In this review, we focus on tumor-derived EVs and their role in the modulation of normal cells infiltrating the sites of solid tumors, as well as on the processes of transformation of a normal microenvironment into a tumor-associated microenvironment mediated by EV crosstalk. We attempt to describe the most significant events in the acquisition of a relatively stable tumor-associated phenotype TME by EVs.

## 2. Cancer-Derived EVs Transform Normal Cells into Cancer Cells

The ability of cancer cell secretions, including EVs, to induce the carcinogenic transformation of normal cells into cancer cells was first experimentally demonstrated, in 1999, by Olmo and colleagues [44]. Subsequently, this hypothesis of a cell-free circulation factor inducing distant tumorigenic transformation was termed “genometastasis”. Early evidence suggested a key role for the horizontal transfer of cell-free DNA into recipient cells [45]. Subsequently, this hypothesis was further elaborated and expanded in [46,47], and tumor-derived EVs have been shown to be the key players in this process [48]. Currently, not all components of EVs have been identified and, more importantly, the mechanism of their malignant effect on normal cells has not been described. Another problem in the studies to date is the difficulty they faced in determining the exact moment of the transformation of normal cells into cancer cells. A change in the phenotype in favor of highly proliferative and apoptosis-resistant cells does not always indicate the ability to develop tumors in vivo. The preferred criteria for identifying cell malignancy could, therefore, relate to in vivo tumor formation upon cell transplantation. The consequence of a transformation of normal cells into cancer cells is an increase in tumor heterogeneity and, accordingly, a decrease in the tumoral response to treatment. Ernesto Yagu et al. showed that the acquired multidrug resistance (MDR) can occur before the malignant transformation stage. The minimal set of changes necessary to obtain pretumorigenic drug-resistant cells includes the expression of telomerase and inactivation of p53 and pRb [49]. A similar alteration might emerge in a normal cell within tumorigenic processes. Indeed, MCF10A normal breast cells transformed with exosomes derived from MCF10A.NeuT tumor cells showed increased resistance to radiation therapy compared to normal cells [50]. A major contribution to the development of tumor EV research was made, in 2008, when Janus Rak’s group introduced the term “oncosomes” and showed their participation in the spread of a mutant form of the epidermal growth factor receptor, termed EGFRvIII, from cancer cells to normal cells [51]. Oncosomes are defined as large vesicles of ectosome pathway biogenesis that facilitate the export of specific oncogenic cargo [52]. Despite the importance of determining the EV subtype, in practice this is a rather difficult task; therefore, most studies are related to common secretory EVs.

One of the most studied components of tumor EVs is RNA. Noncoding circRNA EVs of cancer cells can induce uncontrolled proliferation of normal cells with subsequent transformation into tumor cells [53]. Xiangyu Dai showed that arsenic-induced cancer cells produce exosomes containing circRNA_100284 [54]. When integrated into hepatic epithelial (L-02) cells, circRNA_100284 acts as a sponge of microRNA-217 [55], hybridizes with miRNAs, and inhibits EZH2/cyclin-D1 gene silencing. In another study, it was demonstrated that exosomal hsa_circ_0000069 secreted by SW1990 pancreatic cancer cells promoted the proliferation, migration, and cell cycle progression of human pancreatic duct epithelial cells (HPDEs). These experiments indicate the critical role of exosome-delivered circRNA in tumor progression. Moreover, the malignant changes were reversed through downregulation of hsa_-circ_0000069. When determining the mechanism of action, it was proposed that hsa_circ_0000069 acts as a sponge for miR-144 by increasing the expression of the STIL gene by the main regulator of the mitotic centrosome [56]. In addition to circulating RNAs, it has been shown that gastric cancer cells secrete exosome-packaged long noncoding RNA (*lncRNA*) *SND1-IT1* [57]. Internalization of this lncRNA by human gastric mucosa epithelial GES-1 cells caused cell malignant transformation by enhancing the expression of ubiquitin-specific protease 3 (USP3), inhibiting miR-1245b-5p, and simultaneously recruiting DEAD-box helicase 54 (DDX54). USP3 mediates the deubiquitination of snail family transcriptional repressor 1 (SNAIL1) by activating tumor development and cell transformation [58]. The participation of miRNA packed in EVs in the modulation of the phenotype of recipient cells has been shown in numerous studies, including those on the malignant transformation of normal cells [59,60]. Thus, exosomal miR-224-5p secreted by human colorectal cancer SW620 cells induces cancerous transformation of human normal colon epithelial CCD 841 CoN cells by inhibiting the oncosuppressor CMTM4 [61]. Different types of RNAs are, indeed, powerful tools for modulating cellular processes and creating a heterogeneous population. However, they are not capable of directly causing tumor mutations in the genetic apparatus, which are critical in cell malignancy. We assume that the occurrence of these mutations can be, in part, mediated by indirect mechanisms such as aberrations of DNA repair due to the effects of different types of RNAs.

Our hypothesis is supported by a study of exosomal delivery of annexin A1 (ANXA1) by medullary thyroid carcinoma SW579 cells, which caused the malignant transformation of Nthy ori3 1 thyroid follicular epithelial cell lines that, subsequently, resulted in an increased levels of proliferation, invasion, and epithelial–mesenchymal transition [62]. Recently, Stefanius et al. proposed a two-stage scheme for the involvement of cancer exosomes in the transformation of normal cells into cancer cells (Figure 3) [63].

According to this hypothesis, exosomes secreted by cancer cells, upon being internalized into normal cells, initiate the development of mutations but are not yet able to cause complete transformation into a tumor cell. In this case, exosomes replaced the genotoxic carcinogen 3-MCA (3-methylcholanthrene) that was used as a control and induced random genetic change. During the second stage, the influence of TPA (12-O-tetradecanoylphorbol 13-acetate) enhanced the proliferation in the initiated cells selectively, thus driving malignant transformation of the cells [37,64]. These data are consistent with the studies of Abdour and colleagues, who showed that oncosuppressor genes can work as protectors from the absorption of cancer EVs and, accordingly, from subsequent malignant transformation [64]. In their study, BRCA1 knockout fibroblasts were more susceptible to exosomes isolated from the serum of patients with various tumor locations (colorectal cancer [CRC], hepatocellular carcinoma [HCC], pancreatic cancer [PC], and ovarian cancer) [37]. The treatment led to the malignant transformation of fibroblasts into tumors with the feature of highly proliferative adenocarcinoma. Moreover, the transformed fibroblasts showed signs of a primary tumor: the expression of epithelial markers typical of colorectal adenocarcinoma (CK7, positive for CEA, CK20, CDX-2, and AE1/AE3), markers reflecting early differentiation into pancreatic cancer (positive for cytokeratin CK19 and AE1/AE3, CK7 focal positive patches), and ovarian cancer markers positive for WT-1 and EMA. Interestingly, fibroblasts with wild-type BRCA1 did not transform into cancer cells.

Summarizing the described data, the primary interaction of normal cells with tumor EVs leads to the creation of a heterogeneous population of activated normal cells which, through selection and modification, become either tumor-bearing or cancer-associated or are destroyed by immune surveillance (Figure 4).

In conclusion, cancer-derived EVs can be both initiators and promoters of the malignant transformation of normal cells. In the case of initiatory participation, they introduce nondirectional changes and create conditions for the competitive selection of subpopulations. With promoter involvement, susceptibility to EVs allows for a more directed change in the cell phenotype toward a tumor type. In both cases, there is an accumulation of modifications leading to a malignancy of normal cells. In 2022, Douglas Hanahan updated the list of the main hallmarks of cancer [65]. From this point of view, cancer-derived EVs components are responsible for ensuring the acquisition of these characteristics (Table 2). Unfortunately, not all tumor hallmarks were described for transformed normal cells, but data can be extrapolated from other cells’ types of tumor microenvironment. At the same time, a certain portion of altered normal cells might transform into cancer-associated support cells.

## 3. Cancer-Derived EVs Transform Cancer Cells into Cancer Stem Cells

Cancer stem cells (CSCs), discovered in 1994, consistently attract the attention of researchers [74]. The phenotype of stem cells allows them to self-renew and differentiate into different cell subpopulations. Moreover, the plasticity of CSCs allows them to dedifferentiate back under various stimuli, including those provided by EVs [75]. In addition to the CSC phenotype, because of the tumor heterogeneity, other subpopulations of tumor cells can be detected within the TME [76]. Extracellular vesicles play a significant role in the transfer of an aggressive tumor phenotype to nonaggressive tumor cells, as a result of which the latter acquire the stem cell-like properties [77,78,79].

### 3.1. Support for Cancer Stem Cell Niche by EVs

The features of cancer stem cells require the existence of a stem cell niche that in normal tissues plays an essential role in maintaining stem cells or preventing tumorigenesis by providing primarily inhibitory signals for both proliferation and differentiation. CSC niche characterization, therefore, arises from an intrinsic mutation, leading to self-sufficient cell proliferation, and/or it may also involve the deregulation or alteration of the niche by dominant proliferation-promoting signals. Moreover, tumor development may recruit mechanisms of the niche of normal stem cells for invasion and metastasis [49]. TME cells, particularly cancer-associated fibroblasts, produce EVs that support the population of CSCs and promote the development of tumor chemoresistance. This could be attributed to the enhancement of the Wnt signaling pathway [80,81,82]. Another cell type, glioma-associated human mesenchymal stem cells (GA-hMSCs), has been shown to produce exosomes containing miR-1587, which is known as a tumor-suppressive nuclear receptor corepressor, NCOR1, and leads to increases in the proliferation and clonogenicity of GA-hMSCs according to in vitro experiments [83]. miRNAs also play a significant role in maintaining the stemness of cancer cells [84]. Thus, Yanxia Zhan et al. showed that hypoxic CAF-derived exosomes modulate breast cancer cell stemness through exonic circHIF1A by miR-580-5p cells in breast cancer [85]. In this case, the development of the CSC population can be considered to be a way of adapting the tumor to hypoxic conditions. In general terms, the properties of the cancer stem cell niche are similar to conventional stem cell niches.

### 3.2. EMT-Mediated Transformation of Non-CSCs into CSCs

The induction of non-CSC-to-CSC transformation may be mediated by the epithelial–mesenchymal transition (EMT) [86]. Among various cytokines, it has been demonstrated that TGF-b1 plays an important role in the EMT process. Researchers showed that EVs secreted by chronic myeloid leukemia cells containing TGF-b1 on the membrane trigger the corresponding cascade of events in acceptor cancer cells and promote both the acquisition of the stem cell phenotype and EMT [87]. In addition to the activation of the canonical transcription factor SMAD, signal transduction and activation of the MAPK PI3K/Akt proliferation and survival pathways, like (TGF)-β, PI3K/AKT, and MAPK, were also observed [88]. In addition to activating cancer cell reprogramming of signaling pathways through the activation of receptors, vesicles directly alter the cell transcriptome via the contained transcription factors [89]. In general terms, the reprogramming of cancer cells into cancer stem cells through the EMT has similarities with technology that creates induced stem cells using Yamanaka factors, which include cell pretreatment with TGF-b [90,91].

### 3.3. Role of EV piRNAs in Transformation of Non-CSCs into CSCs

P-element-induced wimpy testis (PIWI) and PIWI-interacting RNA (piRNA) proteins are specific markers for the germline state. They are responsible for maintaining cell stemness [38]. The involvement of piRNAs in the functioning of cancer stem cells was first demonstrated on a testicular germ-cell tumor, where the overexpression of one of the four PIWI protein genes, HIWI, was detected [92]. Since then, the overexpression of piRNAs and PIWI has been found in several other cancers, including breast carcinoma [93,94]. Ross et al., relying on data of increased expression of mobile genetic elements in tumor cells [95], linked the activities of piRNAs and PIWI with the compensatory mechanism of genetic stability. Thus, preadipocyte-derived exosomes have been shown to enhance the growth and survival of breast cancer cells. The exosomal components miR-140/SOX2 and SOX9 were shown to contribute to the maintenance of tumor stemness in that study [96]. Senescent neutrophil exosomes can transfer chemoresistance and EMT characteristics to recipient breast cancer cells through cell-to-cell transfer of piR-17560. Moreover, exosomal piR-17560 promotes the EMT by regulating ZEB1 of breast cancer cells through FTO-dependent m6A demethylation. These findings indicate the critical role of senescent neutrophils in the regulation of the EMT and, consequently, the acquisition of the CSC phenotype [97]. piR-823 also had different activities in various types of cancer. For example, piR-823 promotes cell proliferation and tumorigenesis by enhancing HSF1 phosphorylation and transcriptional activity in colorectal cancer [98]. In multiple myeloma, piR-823 promotes tumorigenesis by regulating de novo DNA methylation [99]. However, piR-823 has been shown to inhibit gastric carcinogenesis [100] and induces the expression of stem cell markers (including OCT4, SOX2, KLF4, NANOG, and hTERT) and increases the formation of mammospheres, as has been demonstrated on breast cancer MCF-7 and T-47D cells. The authors attribute this to the activation of the Wnt signaling pathway through increased expression of DNA methyltransferase (DNMT) and methylation of the adenomatous polyposis coli (APC) gene [101]. The induction of the overexpression of piR-823 has not yet been studied. It has recently been shown that piR-823 can be secreted by cancer cells via exosomes, and it was found in biopsies from patients with colorectal cancer [102]. Extrapolating from these data, piR-823 secreted by CSC during tumor development may be involved in the autocrine and paracrine maintenance of the stem phenotype in the cancer population. It also creates a favorable environment, as shown in a multiple myeloma model, whereby exosomal piR-823 induced the transformation of EA.hy926 endothelial cells by enhancing the expression of VEGF, IL-6, and ICAM-1, and attenuating apoptosis [49].

### 3.4. Role of Other EV Cargoes in the Transformation of Non-CSCs into CSCs

Another important molecule for CSC maintenance is Wip1 (wild-type p53-inducible phosphatase-1) [103]. Indeed, the inhibitions of p53 and p38 MAPK pathway activities have been found to result in an extended MSC life span and their increased differential potential in normal mesenchymal stem cells [104]. A similar role for Wip1 has been identified in cancer stem cells, in which it is able to directly dephosphorylate p53 at Ser15 and dephosphorylate MDM2, an E3 p53 ubiquitin ligase, leading to the destabilization of the p53 protein. Wip1 also dephosphorylates and inactivates ATM, Chk1, and Chk2, which are upstream activators of p53 kinase [105,106]. The inhibition of the p38 MAPK pathway is also exerted via the Wip1 phosphatase activity [107]. So far, there are no data on the exosomal secretion of Wip1 by cancer cells, although it is known that changes in Wip1 expression in cells can occur under the influence of exosomal circSHKBP1 secreted by cancer cells [108].

In general terms, the transformation of cancer cells into CSCs and maintenance of the stem phenotype by EVs are similar to the creation of a stem niche [109]. The questions of why a CSC population is formed in a tumor and why it is so relatively small still remain open. The accumulated data clearly indicate that CSCs increase tumor resistance to therapy [110]. Indeed, the CSC population is heterogeneous and, during treatment-related selection, relapses quickly because of high proliferative activity and plasticity [111]. As was shown by Bao et al., glioma stem cells (defined by CD133 expression) contributed to tumor radioresistance through the activation of the DNA damage checkpoint response, which was reversed by the application of specific inhibitors of checkpoint kinases (Chk1 and Chk2) [112]. Another study showed that the BMP pathway in glioma stem cells is related to cell resistance to not only radiotherapy but also to chemotherapy with temozolomide [113].

## 4. Cancer-Derived EVs Transform Stromal Microenvironment Cells into Cancer-Associated Support Cells

The cellular composition of the TME, on the one hand, is quite well studied, but on the other hand, in practice, determining the cell type is a rather difficult task due to high heterogeneity and necrotic processes [105]. Most data are available regarding cancer-associated fibroblasts (CAFs), tumor-associated macrophages (TAMs), tumor-associated neutrophils (TANs), and tumor-infiltrated/-associated natural cells (TINKs/TANKs) (Table 3). The remaining cellular components of the TME, however, have been studied to a lesser extent. The transformative influence of cancer-derived EVs on the formation of cancer-associated support cells with certain phenotypes is shown in Table 3.

### 4.1. Transformation of Fibroblasts into Cancer-Associated Fibroblasts (CAFs)

Fibroblasts are connective tissue cells of mesenchymal origin [131]. They are often identified by their morphology, location, and absence of clonal markers of endothelial cells, epithelial cells, and leukocytes. However, their unique molecular markers have not yet been identified. Vimentin and platelet-derived growth factor receptor-α (PDGFRα) can be used as auxiliary markers [12]. Cancer-associated fibroblasts functionally differ from normal ones by their more pronounced tumor-modulating effect, expressed in the induction of angiogenesis, inflammation, and remodulation of the extracellular matrix (ECM) [132]. The identification of CAFs has typically been carried out on the basis of the expressions of various “CAF markers”, such as fibroblast activation protein alpha (FAP) and alpha smooth muscle actin (αSMA). However, αSMA can also be highly expressed in normal fibroblasts when wound healing occurs [13]. Moreover, αSMA is constitutively expressed in smooth muscle cells, as is FAP in adipocytes and osteocytes [133,134]. The high heterogeneity of CAFs is reflected by their various subpopulations within a tumor [135]. It has been found that subpopulations of inflammatory fibroblasts (iCAFs) and myofibroblasts (myCAFs) in PDACs are spatially separated. myCAFs are located in the periglandular region, whereas iCAFs (characterized by the secretion of IL6, IL11, and LIF and a stimulated STAT pathway) are distantly located from cancer cells and myCAFs [136]. The main distinguishing feature and boundary of the transition between CAFs and tumor cells is the absence of genetic mutations. The identification of CAFs is usually made on the basis of many traits, including molecular markers, morphologies, and genotypes [132]. Depending on the expression of CAF markers (i.e., FAP, αSMA, and integrin β1 [CD29]), various CAF subsets were previously identified for ovarian and breast cancers [137,138,139]. The myCAF subsets (FAP^High^ SMA^High^ CD29^Med-High^ and FAP^Neg^ SMA^High^ CD29^High^) have been shown to be correlated with a poor prognosis [140,141]. The process by which fibroblasts are converted to CAFs is not fully understood. However, various molecules and biochemical processes, including inflammatory cytokines (IL-1, IL-6, TNF, and TGFβ), RTK ligands (PDGF and FGF), physiological stress (e.g., ROS and disrupted metabolism), and DNA damage, have been shown to play a role in CAF formation [142,143,144,145]. The stiffness and composition of the ECM, as well as contact signals (i.e., Notch and Eph/ephrins), are also important for fibroblasts’ transformation [146]. The particular role of EVs in this process of CAF formation is rather difficult to estimate because of the additive effect of the multiple factors listed above. However, one cannot underestimate the significance of cancer-derived EVs in the transformation of fibroblasts into CAFs. CSCs play an important role in this process through their involvement in modulating the microenvironment [147]. The main effect of CSCs produced by EVs is to enhance fibroblasts’ proliferation, migration, and invasion [148,149,150]. Thus, it has been shown that Piwil2-induced cancer stem cells (Piwil2-iCSCs) increase the expressions of MMP2 and MMP9, which are responsible for enhanced invasiveness and migration through exosomes [114]. It has been shown in melanoma models that exosomes secreted by B16F0 cells induce reprogramming of NIH/3T3 fibroblast cells to CAFs, as evidenced by increased expressions of CAF-related markers (α-SMA and FAP) and facilitation of cell migration. EVs secreted by B16F0 melanoma cells deliver Gm26809 to NIH/3T3 cells, where Gm26809 (long noncoding RNA [lncRNA]) mediates the reprogramming of normal fibroblasts. As expected, knockdown of the Gm26809 gene disrupts exosome-induced transformation [115]. Other miRNAs within cancer-derived exosomes can also induce transformation [144]. Thus, exosomes of breast cancer cells containing miR-146a inhibited the expression of the TXNIP gene in fibroblast cells and activated the transition to CAFs via activation of the Wnt signaling pathway [116]. Another agent, miR-27a, found in abundance in exosomes of gastric cancer, is able to directly target the oncosuppressor CSRP2 by binding to its 3’-UTR, thereby triggering Rac1 activation and inductions of the ERK and PAK/LIMK/cortactin signaling cascades [117,118]. Gutkin et al. determined that hTERT mRNA, a transcript of the telomerase enzyme, is transferred from cancer cells through exosomes to telomerase-negative fibroblasts, where it is translated into a fully active enzyme. Subsequently, these telomerase-positive cells represent a new cell type—nontumor cells with enhanced telomerase activity [119]. A more recent study reported that exosomal telomerase may play a role in modifying normal fibroblasts into cancer-associated fibroblasts by upregulating αSMA and vimentin [68]. As previously mentioned, the interaction of tumor cells with CAFs is mutualistic. In addition to the obvious benefits for the tumor in the form of increased angiogenesis, extracellular matrix remodeling, and activation of the inflammatory processes, it has been shown that CAFs with exosomal lncRNA POU3F3 induced tumor resistance to cisplatin due to the secretion of inflammatory cytokines [151]. Accordingly, they cause an increase in tumor resistance to therapy [152]. However, the involvement of CAFs in tumor development shows a dual role. In a study by Daniela et al., two subpopulations were identified, as follows: CAF-N with a transcriptome and secretome similar to normal fibroblasts, and tumor-promoting CAF-D with an expression pattern that differed from normal fibroblasts and CAF-N [153]. In addition, CAFs expressing FSP1 have been shown to inhibit tumor development by encapsulating carcinogenesis. When the carcinogen methylcholanthrene (MCA) was administered subcutaneously, a concentration of FSP1-positive fibroblasts around the lesion and increased secretion of collagens were observed [154]. Another marker of antitumor CAFs is the glycosylphosphatidylinositol-anchored protein Meflin, the increased expression of which induces an improvement in chemosensitivity in PDAC [155]. Complete depletion of αSM-positive CAFs in the tumor PDAC model in a transgene murine model resulted in reduced desmoplasia and stromal stiffness and increased tumor invasiveness and metastatic activity [156]. The idea of the existence of protumorogenic and antitumorogenic types of CAFs emerged after a series of failures associated with CAF depletion [157]. Today, it is considered that the most promising anticancer strategy mediated by CAFs is to switch CAFs to a quiescent state [158]. Silke Haubeiss et al. determined the therapeutic role of dasatinib mediated through the transformation of CAFs into normal resting fibroblasts; moreover, the incubation of tumor cells with conditioned medium from CAFs preincubated with dasatinib significantly reduced tumor cell proliferation [159].

Unfortunately, there are no data on the dynamic changes in CAFs during cancer progression. According to the previously reported studies, CAF senescence occurs over several passages in vitro and depends on the age of the tissue [131]. When drawing analogies between the processes of transformation of fibroblasts into CAFs and wound healing (in both cases, activation of fibroblasts and acquisition of the myofibroblast phenotype are observed), the role of tumor EVs consists in the maintenance of a chronic activated state of CAFs [160]. Unfortunately, there are no experimental data on the dynamics of the changes in the CAF phenotype and reversible tumor effects. Targeted uptake of EVs has been documented as therapeutic [161]. Furthermore, it has been previously shown that tumor EVs can carry PDGF on their surfaces [162], and PDGFRα/β is one of the markers of CAFs [12]. This suggests the existence of the targeted secretion of EVs by tumor cells for CAFs. However, the question of whether this is the case remains open and requires in-depth study.

### 4.2. Transformation of Nerve Cells into Cancer-Associated Nerve Cells

Tumor innervation, along with angiogenesis, is a necessary process for disease progression [163]. Phenotypic transformations of nerve cells under the influence of tumors can be divided into the following three aspects of innervation: axonogenesis, reprogramming of nerve cells, and neurogenesis (Figure 5) [164]. Nerve cells with a phenotype that changed under the influence of tumor cells do not have a specific classification, unlike fibroblasts or macrophages. Identification is mainly provided by histological staining and morphology. As candidate molecular markers, high expressions of Eph receptors as the main stimuli of innervation could potentially be used in the future [165]. The main task solved by these aspects of innervation is to increase the supply of the tumor with various neurotransmitters. In a prostate cancer model, it has been shown that norepinephrine (NE) delivered from nerve terminals acts on b2- and b3-adrenergic receptors (Adrb2, Adrb) expressed on stromal cells, promoting the survival of cancer cells and the initial development of the tumor. Nerve fibers from the parasympathetic nervous system (PNS) also invade tumors, delivering acetylcholine (Ach), which promotes tumor cell proliferation and egress to lymph nodes and distant organs through the type 1 muscarinic receptor (Chrm1) expressed on stromal cells [166].

Axonogenesis is among the aspects of tumor innervation [27]. The intensity of this process correlates with a poor treatment prognosis and the rate of tumor progression. It is not completely known what mechanisms underlie the connectivity of these processes. It is known that the extracellular release of neurotrophic factors, such as nerve growth factor [168], by tumor cells can promote cancer development [169]. Similarly, NGF is synthesized in tumor cells in the form of proNGF. Following intracellular internalization with subsequent furin protease cleavage or after extracellular procession by metalloproteases, proNGF is transformed to its matured form [170]. One of the main roles in the induction of tumor-associated axonogenesis is exerted by cancer-derived EVs. Madeo et al. determined that exosamal EphrinB1 induces and activates axonal outgrowth in tumors, enhancing tumor proliferation [28]. In addition, plasma-derived exosomal EphrinB1 isolated from human head and neck cancer specimens and mouse oropharyngeal squamous cell carcinoma caused increased neurite outgrowth in rat pheochromocytoma PC12 cells compared to exosomes derived from control plasma of healthy donors [171]. The involvement of exosomal miRNAs was shown in a study by Amit et al., in which the authors showed that genetically aberrant, p53-knockout or -mutant (p53C176F and p53A161S) oral squamous cell carcinoma (OCSCC) cells release EVs that promote neuritogenesis in the posterior ganglion roots. The effect was attributed to the miR-34a, miR-21, and miR-324 contained in exosomes [172].

Neurogenesis, as a process of de novo formation of neurons, is extremely difficult to study, especially within the framework of tumorigenesis. However, Mauffrey and colleagues expounded on this hypothesis, demonstrating that neural progenitor cells expressing the neural stem cell marker doublecortin (DCXþ) migrate from neurogenic regions of the brain’s subventricular zone (SVZ) to tumorous and metastatic niches via the bloodstream, differentiating into noradrenergic and mature neuronal phenotypes [173]. In addition, the acquisition of a neuron-like phenotype by cancer stem cells has also been demonstrated. These cells expressed autonomic nerve markers such as VAChT (a marker of parasympathetic neurons) or TH (tyrosine hydroxylase, which is characteristic of sympathetic neurons) [174].

To a particular extent, the transformation of nerve cells under the influence of tumors is relevant for the human brain environment, which is distinguished by a wide variety of cells, including astrocytes, macrophages, vascular endothelial cells, microglia, macrophages, and fibroblasts [175]. It is known that the development of a tumor near the nerve pathways is mediated by the growth in cancer cells around the peripheral nerves and, as a result, by intrusion into those nerves; this process is termed perineural invasion (PNI). In the case of tumors of the central nervous system, such a variant of development is unavoidable. Research in this area received further impetus with the discovery of close interactions between tumor cells and brain cells mediated by long membranous protrusions termed tumor microtubes (TMs) [176]. The most obvious factor in the transformation of neurons in CNS tumors is solid stress developed during the promotion of cell density, which can cause nonreversible and reversible nerve dysfunction [177]. The role of EVs in these processes is reduced to the formation of a pro-oncogenic microenvironment, often mediated by neuroinflammation. EVs released by glioblastoma cells induce migration and secretion of growth and pro-inflammatory factors in astrocytes. In addition, the regulation of the transcription factors TP53 and MYC by EVs induced the appearance of a senescence-associated secretory phenotype (SASP) in astrocytes [178].

### 4.3. Transformation of Immune Cells into Cancer-Associated Types

Immunological surveillance is one of the first barriers to tumor occurrence. However, with further tumor progression, immune cells can become supportive of tumor growth. A prime example is macrophages, in which the tumor-associated macrophage (TAM) phenotype has been identified [179]. TAMs are identified as protumor members of the microenvironment through immunohistochemical staining of neoplastic tissue for CD68 [180]. However, it has also been determined that the TAM population is heterogeneous and includes both M1- and M2-activated macrophages, exhibiting either protumorigenic or antitumorigenic effects, respectively. The overall effect of the TAM environment on the tumor is considered in the “macrophage balance hypothesis” paradigm [181]. Four approaches are commonly used to characterize macrophage subpopulations: analysis of cell surface marker expression, expression of transcription factors, production of cytokines, and production of specific enzymes [182]. The role of each population is still not defined, but they all been found to have a close relationship with the tumor cells and their microenvironment [183].

M2 polarization of macrophages is the main event associated with the formation of TAM and is characterized by the secretion of anti-inflammatory factors and pro-oncogenic activity [184]. It has been demonstrated that this process is mediated by EVs that contain certain cytokines (IL-6, TGF-b, IFN-a, and others) in various types of tumors [185]. In addition, M2 polarization is mediated by a vast ensemble of different RNAs contained in EVs [168]. M1 polarization is often associated with the secretion of pro-inflammatory factors and, accordingly, antitumor activity. The EVs of such macrophages enhance the therapeutic effect of, for example, paclitaxel [186]. The reprogramming of M1 macrophages into M2 and, accordingly, into TAMs can be carried out by EVs [168]. This process is similar to the natural processes that occur with macrophages during wound healing [187]. Reverse transformation of M2 macrophages into M1 is a promising strategy to enhance anticancer therapy [188]. Nonetheless, an increase in the TAM population in a tumor cannot only be facilitated by the EVs of tumor cells. For example, it has been shown that EVs secreted by human and mouse MSCs accelerated breast cancer progression by inducing differentiation of monocytic myeloid-derived suppressor cells (MDSCs) into highly immunosuppressive M2-polarized macrophages. The EVs of MSCs, in contrast to the EVs of tumor cells, contained TGF-β, C1q, and semaphorins, which induce the overexpression of PD-L1 in both immature myelo-monocytic precursors and committed CD206+ macrophages and, thereby, increase overall immune tolerogenicity [189].

The tumor-associated neutrophil (TAN) is another immune participant in the tumor microenvironment. In analogy with macrophages, a subpopulation of transformed neutrophils with protumor functions is termed N2 TAN [16]. Researchers attribute a central role in this process to TGF-b, the expression of which is controlled not only by the cytokine components of exosomes but also by miRNA [190]. In addition to this, an important exosomal component of this process is high-mobility group box-1 (HMGB1), which activates the NF-κB pathway through interaction with TLR4, resulting in an increased autophagic response in neutrophils [124]. The main antitumor effects of the N1 TAN population are associated with the activation of T cells, induction of apoptosis of tumor cells (via TRAIL), and production and secretion of ROS, as well as participation in ADCC [16]. Using a model of TAN transformation, it has been shown that the acquisition of a cancer-associated phenotype not only supports the tumor progression but also prolongs the life cycle of neutrophils through the induction of resistance to apoptosis [191].

Natural killer (NK) cells are another type of leukocyte capable of infiltrating tumors and acquiring a tumor-associated phenotype (TANKs) found in peripheral blood [18]. Like other immune cells, they exhibit a dual role in the TME [192]. They have been shown to play a role in enhancing angiogenesis and tumor invasion. It has also been demonstrated that a high content of TINKs in the TME in primary head and neck squamous cell carcinoma (HNSCC) correlates with a good prognosis [11,193]. TINKs have been characterized as CD56 bright, and they overexpress NKG2A, as well as lower levels of KIRs and leukocyte immunoglobulin-like receptor subfamily B member 1 (LILRB1) [19]. Tumor EVs primarily have an immunosuppressive effect on NK cells [194,195]. The main exosomal component responsible for reducing the cytotoxicity of natural killer cells is TGF-β1, which causes a decrease in the expression of activating receptors, including NKG2D, NKP30, NKP44, NPK46, and NKG2C [126]. It is noteworthy that the tumor mechanisms of the inhibition of NK activity through the release of NKG2DL are similar to the processes of embryonic development, when the placenta secretes NKG2DL to protect the fetus from the cytotoxic effects of NK [127,196]

Immune cells often assemble into tertiary lymphoid structures (TLSs) [197]. TLSs are organized aggregates of immune cells that form in nonlymphoid tissues associated with autoimmune reaction, chronic infection, cancer, and other diseases in the postnatal period characterized by an inner zone of CD20+ B cells that is surrounded by CD3+ T cells [198]. The molecular mechanisms by which tumor-derived EVs influence TLS formation remain unknown [199]. Thus, EVs are involved in the formation of TLSs at the initiation and maturation stages, and apoptotic bodies play a key role in these processes [200]. The function of TLSs in tumor progression remains unclear (Figure 6). On the one hand, TLSs duplicate the role of SLO and are involved in enhancing the immune response to cancer cells [201]. It has been shown that in primary melanoma, a high density of DC-Lamp+, a mature DC found within lymphoid aggregates, is associated with a strong infiltration of activated T cells and significantly higher rates of disease-free survival [202]. The anti- or protumorigenic role of DC is defined by its cell maturity status [203]. Immature DCs that have infiltrated into the tumor display low expression of costimulatory molecules (CD80 and CD86) and high expression of inhibitory molecules (PD-L1 and CTLA-4), and, therefore, create immunotolerogenic conditions within the TME [204]. Meanwhile, mature DCs in the TME display antitumor immune responses through the release of IFN-λ1, which stimulates Th1 differentiation and activation and effector CD8+ T-cell activation together with enhanced IFN-γ production through IL-12p70 production, which, as a result, increases overall survival [205]. On the other hand, a correlation has been found between late-stage progression of breast, bladder, or stomach cancer and the number of TLSs formed [206,207,208]. Clearly, the participation of TLSs in tumor progression depends, first of all, on cellular components [209] such as subpopulations of dendritic cells [210], T lymphocytes [211], and B lymphocytes [30,31]. Whether the effect of cancer EVs on TLS formation is inhibitory or enhancing is unknown. Despite this, TLSs remain promising prognostic markers and a potential target for therapy [197].

The remaining fractions of immune cells are much less studied and phenotypically identified in the light of tumor transformation, which is due to the lower contents of these cells in the tumor microenvironment [213].

### 4.4. Cancer-Derived EVs for Transformation of Other Cell Types

In addition to the previously described cells, the tumor microenvironment includes a number of other tumor-supporting cell types, including endothelial cells, adipocytes, and keratinocytes [214].

Endotheliocytes are the main cells of blood vessels and one of the main participants in angiogenesis, which, in turn, is targeted by many secreted factors including EVs [215]. Tumor interactions with endothelial cells induce their transformation into tumor endothelial cells (TECs) characterized morphologically by irregular surfaces, excessively fenestrated cell walls, and loose intercellular junctions to adjacent cells. They exhibit a stem-cell-like origin, thus playing a key role in tumor neo-angiogenesis [22]. The activation of VEGF signals released by tumor cells and the TME plays a central role in this process [216]. In addition, tumors have been shown to release exosomes that inhibit the mechanosensitive ion channel transient receptor vanilloid 4 (TRPV4), whose expression and activity is significantly reduced in tumor endothelial cells (TECs). The activation of TRPV4 has been shown to normalize the tumor vasculature and improved the efficacy of anticancer therapy [129].

Adipocytes are cells involved in energy metabolism. They are found in large numbers in the mammary glands and are therefore active participants in the TME in breast cancer [217]. A recently described phenotype of cancer-associated adipocytes (CAAs) (characterized by the production of CCL5, CCL2, IL-6, and TNF-α) has been shown to promote the proliferation and invasion of tumor cells, as well as neovascularization [25]. One of its key roles in CAA activation is attributed to exosomal miR-1304, which regulates GATA2 gene expression and enhances lipid release from adipocytes [130]. Subsequently, Munteanu et al. proposed treatment in other tumor models based on the downregulation of CAAs [218].

Keratinocytes, cells that are mainly located in the epidermis of the skin, are constitutive participants in the TME in various forms of melanoma [219]. In one study, Danella et al. reported that cancer-associated keratinocytes in a head and neck squamous cell carcinoma (HNSCC) model secreted TGFβ and TNFα, which enhanced the invasion of HNSCC cells [26].

Pericytes, together with smooth muscle cells, are mural cells serving to support and stabilize the endothelium. This explains their significant role in the processes of neoangiogenesis [220]. In competition with pericytes, cancer cells replace them and line blood vessels. In this case, detached pericytes undergo pericyte–fibroblast transition (PFT), which facilitates vascular penetration and metastasis of cancer cells [221]. Xiaofei Ning et al. showed that an exosome from gastric cancer induced the transformation of pericytes via the activation of the PI3K/AKT and MEK/ERK signaling pathways. On the contrary, the inhibition of the BMP pathway reversed cancer exosome-induced CAF transition [222].

Lymphatic endothelial cells (LECs) constitute the main cell type in lymph nodes and vessels. A major breakthrough in the study of lymphatic endothelial cells occurred with the discovery of their specific markers: 5’-nucleotidase, lymphatic vessel endothelial receptor-1, vascular endothelial growth factor receptor-3, podoplanin, and Prox-1 [223]. It was subsequently demonstrated that LECs are an integral part of the TME involved in processes of lymph metastasis. Phenotypic changes in LECs within the tumor supported the identification of structures called tumor-draining lymph nodes [32]. This resulted in the identification of the following processes: (1) lymphangiogenesis and the expansionof lymphatic sinuses, (2) dilation and dedifferentiation of high endothelial venules (HEVs), and (3) remodeling of fibroblastic reticular cells (FRCs) [224].

## 5. Therapeutic Applications of EVs Involved in TME Modulation

Close interactions of TME cellular components can be utilized in tumor diagnostics and therapy. Indeed EVs, as carriers of various molecules (mRNA, functional proteins, etc.), were shown to provide a real-time valuable biomarker platform. From what was described earlier, it is also clear that the role of EVs within TME interaction is considerable. The use of artificial target exosomes for tumor therapy is a well-studied strategy [225]. Such therapeutic exosomes have a number of advantages, such as biocompatibility and biodegradation, low toxicity, high stability, high permeability through cellular barriers, high specificity, accumulation in tumor tissues, and the potential for design [5]. Engineered EVs often target tumor cells directly, but the targeting of other TME cells for reprogramming has recently received a new impetus (Table 4).

As previously described, this strategy of cell reprogramming, compared with complete depletion using the example of CAFs, shows great promise [156]. Targeted reprogramming of individual TME cell types makes it possible to alter the composition of a solid tumor [229]. This, in turn, allows greater tumor susceptibility to chemo- and radiotherapy, immunotherapy, and other treatment strategies [233]. Unfortunately, EVs have currently only been developed for certain types of TME cells. Additionally, the arsenal of therapeutic agents is quite limited. We hope that increased understanding of cellular cross-talk within the TME will help to expand this tumor treatment strategy.

The application of EVs in TME modulation is not, however, limited to the depletion and reprogramming of cellular components. Modified exosomes with multivalent antibodies on the surface specific to T-cell CD3 and cancer-cell-associated EGFR redirect and activate cytotoxic T cells toward cancer cells for killing [234]. In addition, therapeutic exosomes aimed at metabolic reprogramming of the TME deserve special attention [235]. Furthermore, strategies are being developed in parallel to inhibit either the release of EVs or their uptake by TME cells [236]. The development of exosome-based vaccines deserves special attention, as reviewed in [237,238]. The clinical application of EVs in the treatment of tumors is becoming an increasingly promising tool. Many synthetic, natural, and combined EVs are undergoing clinical trials (Table 5). We believe that the development of exosome-based tumor therapy will be associated with the construction of more complex networks of interactions between TME components and more targeted effects. There have already been studies linking tumor gene therapy with subsequent TME remodeling induced by primary exposure [239].

## 6. Conclusions

Although great strides have been made in recent decades in the study of the transformation of the tumor microenvironment under the influence of cancer cells (mediated by EVs), specific mechanisms still remain poorly understood. The scientific research carried out to date has mainly focused on the study of a binary system, in which, on the one hand, there is a cancer cell, and on the other, a normal cell of the body, which undergoes certain changes under the action of tumor molecules. However, this system does not take into account the influences of various transformed cells of the tumor microenvironment (for example, endotheliocytes, CAFs, pericytes, and TAMs) on each other, which undoubtedly determine the progression of the tumor, its heterogeneity, and resistance to therapy. It is presumably these complex intercellular interactions that determine the occurrence of metastatic niches and the formation of microemboli (conglomerates of normal and cancer cells) [240]. Accordingly, such metastatic niche formation was shown to be mediated by EVs during the processes of apoptosis and senescence of CAFs due to the abundant release of inflammatory and growth factors [241,242]. In these aspects, TME may be regarded as an ecosystem, in which EVs play a significant role in the systemic tumor–host interplay, modulating interactions among cancer cells and their local microenvironment, distant organ niches, and nervous, endocrine, and immune systems [243,244].

It is also worth noting that cancer cells successfully hijack various physiological processes (wound healing and stem niche formation) in the body for cell recruitment and remodeling [245]. This may be due to the imitation of other pathological processes by the TME. Further study of the tumor-derived EVs may help to elucidate the complex underlying pathways of normal cell recruitment to the tumor site and subsequent transformation. The presence of many tumor-associated phenotypes of normal cells in a tumor site is associated with a complex biocenotic system. At the same time, EV crosstalk is one of the pathways of transformation of normal competitive cells into supporting tumor-associated cells.

Tumor-derived EVs have shown potential to be employed not only as detectable biomarkers for early diagnosis but also as a promising tool for targeted therapeutic strategies. The accumulated data clearly indicate that novel therapeutic strategies should include targeting the TME, which, in turn, may sever tumor resistance mechanisms and increase the efficacy of applied therapies.

## Figures and Tables

**Figure 1 cells-13-00682-f001:**
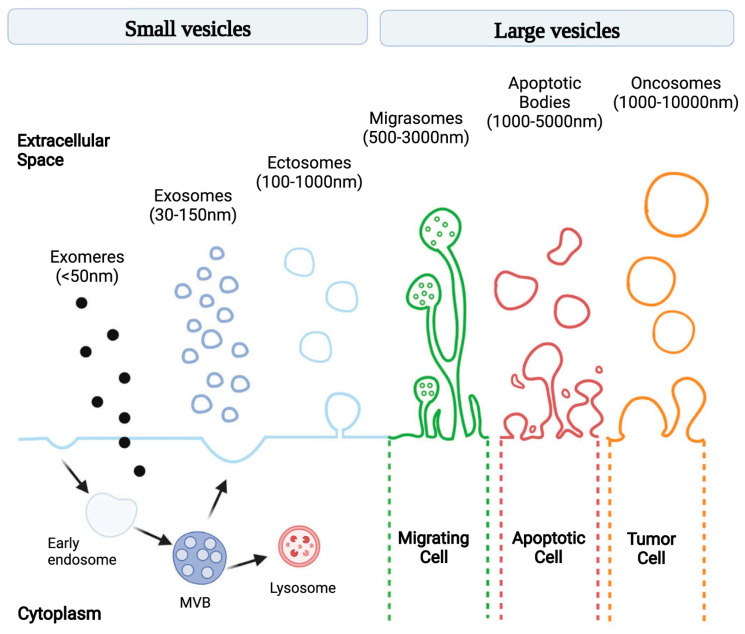
Schematic illustration of EV subtypes according to their different size and genesis models [3].

**Figure 2 cells-13-00682-f002:**
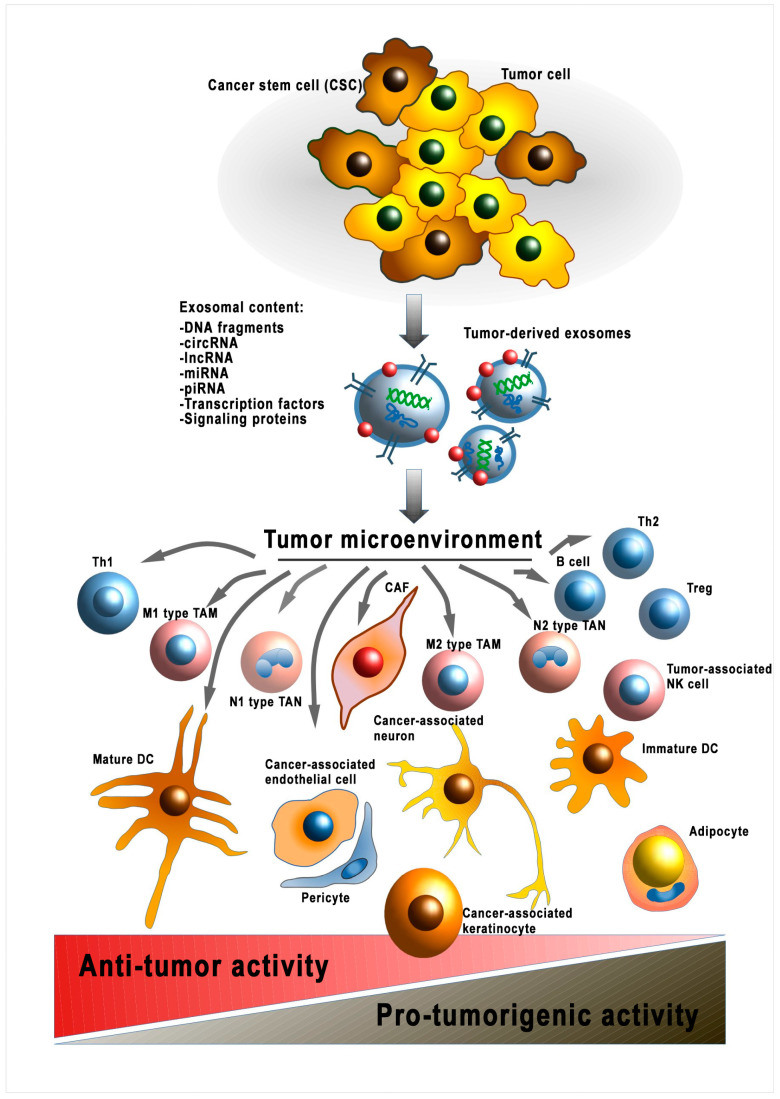
Modulation of the tumor microenvironment by cancer-cell-derived EVs. sRNA—ribonucleic acid; miRNA—micro RNA; lncPRN—long noncoding RNA; circRNA—circular RNA; piRNA—piwi-interacting RNA; DNA—deoxyribonucleic acid; Th1—T-helper 1; TAM—tumor-associated macrophage; Treg—T regulatory cell; TAN—tumor-associated neutrophil; CAF—cancer-associated fibroblast; DC—dendritic cell; Th2—T-helper 2; NK cell—natural killer cell.

**Figure 3 cells-13-00682-f003:**
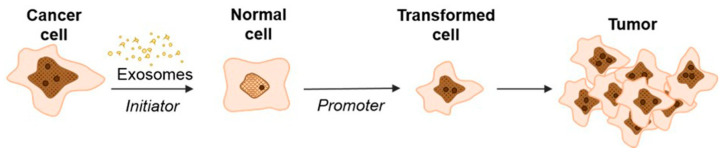
Schematic model of exosome-mediated transformation [63].

**Figure 4 cells-13-00682-f004:**
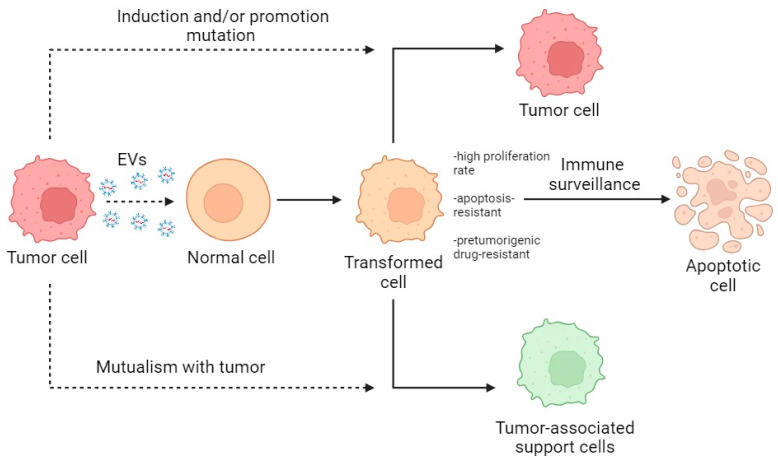
Scheme of the transformation of normal cells within the TME. Dotted lines indicate the direction of action; solid lines indicate the direction of transformation.

**Figure 5 cells-13-00682-f005:**
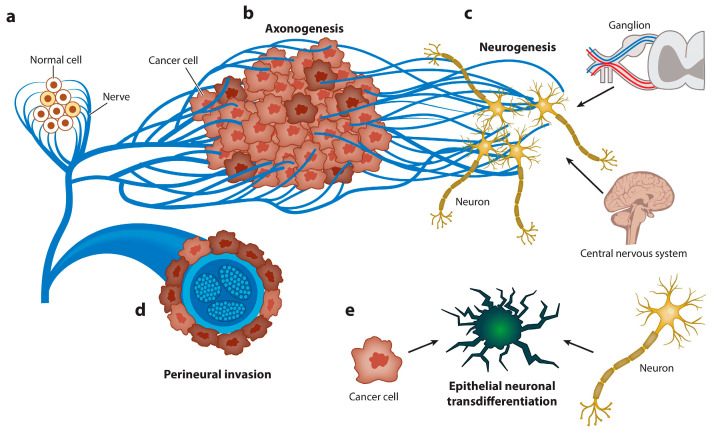
Graphical representation of the spectrum of neuroepithelial interactions. Cancer recapitulates the biology of the neural regulation of epithelial tissues. (**a**) Nerves regulate the homeostasis and energetic metabolism of normal epithelial cells. (**b**) Axonogenesis from pre-existing nerves takes place to supply the new malignant epithelial growth. Cancer cells rarely develop in denervated organs, and denervation affects tumorigenesis in vivo and in humans. (**c**) Neurogenesis occurs later, first in ganglia around organs or the spinal column and, subsequently, through recruitment of neuroblasts from the central nervous system. The hallmark of this stage is the regulation of homeostasis and energetic metabolism. (**d**) Perineural invasion is the most efficient interaction between cancer cells and nerves. The hallmarks of this stage are increased proliferation and decreased apoptosis. (**e**) Finally, carcinoma cells transdifferentiate into a neuronal profile to gain neural independence [167].

**Figure 6 cells-13-00682-f006:**
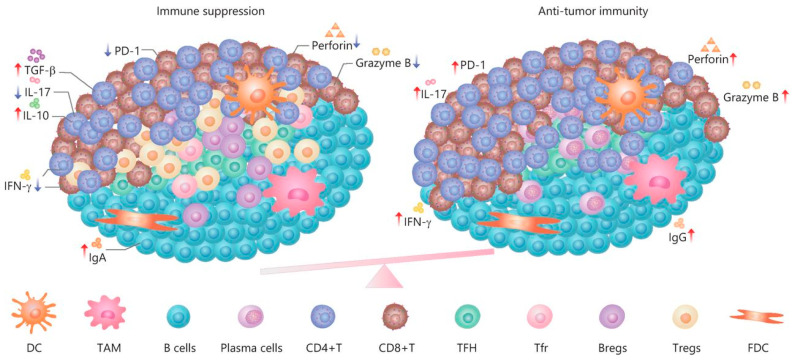
The dual role and immune regulation in TLSs [212].

**Table 1 cells-13-00682-t001:** Properties of tumor microenvironment cells associated with protumorigenic effect.

Cell Type	Morphological Characteristics	Pathophysiological Characteristics	Molecular Profile	Reference
Cancer-associated fibroblasts (CAFs)	Loss of fusiform form	-High secretion of ECM components -Tumor progression	αSMA, FAP, vimentin, and PDGFRα	[12,13]
Tumor-associated macrophages (TAMs)	Often elongated morphology, increase in size	-Immunosuppressive activity -Tumor progression	M2-type polarization CD163, CD206, Arg1, and IL10	[14,15]
Tumor-associated neutrophils (TANs)	Not defined	-Immunosuppressive activity -Tumor progression	Overexpression of chemokines CCL2, CCL3, CCL4, CCL8, CCL12, CCL17, CXCL1, CXCL2, IL-8/CXCL8, and CXCL16	[16,17]
Tumor-infiltrated/-associated natural killer cells (TINKs/TANKs)	Not defined	-Immunosuppressive activity --Tumor progression	CD56bright, overexpressed NKG2A, and lower levels of KIRs and LILRB1	[18,19]
Tumor-infiltrated/-associated dendritic cells (TIDCs/TADCs)	Form tertiary lymphoid structures (TLSs)	-Immunosuppressive activity -Tumor progression -Do not present tumor-derived antigens	Not defined	[20,21]
Tumor endothelial cells (TECs)	Irregular surfaces, excessively fenestrated cell walls, loose intercellular junctions to adjacent cells	Tumor progression	Aneuploidy, VEGF autocrine loop, responsiveness to EGF, and adrenomedullin	[22,23]
Cancer-associated adipocytes (CAAs)	Elongated form, loss of a considerable number of lipid droplets	Tumor progression	Overexpressed CCL5 and CCL2, increased production of the pro-inflammatory cytokines IL-6 and TNF-α	[24,25]
Cancer-associated keratinocytes (CAKs)	Not defined	Tumor progression	DMBT1 suppression	[26]
Cell type without certain phenotype				
Nerve cells	Increase in axonal network	Tumor progression	Overexpressed Eph	[27,28]
Pericytes	Loss of integrity with the blood vessel	Tumor progression	Pericyte–fibroblast transition (PFT)	[25]
T lymphocytes	Form tertiary lymphoid structures (TLSs)	Secretion of pro-inflammatory cytokines and chemokines	Not defined	[29]
B lymphocytes	Form tertiary lymphoid structures (TLSs)	Secretion of immunosuppressive cytokines and antibodies	Secretion of IL-10, TGFb, and IgA	[30,31]
Lymphatic endothelial cells (LECs)	Recruit myeloid lymphatic endothelial cell progenitors (M-LECPs) from bone marrow for lymphoangiogenesis	Tumor progression	LYVE1 and podoplanin	[32]

**Table 2 cells-13-00682-t002:** Influence of cancer-derived EVs on the acquisition of tumor hallmarks in transformed normal cells.

Hallmark of Transformed Cancer Cell	Cancer-Derived EV Factors	Mechanism of Action	Reference
Sustaining proliferative signaling	H-ras and K-ras transcripts; miR-125b, miR-130b, and miR-155; Ras superfamily of GTPases Rab1a, Rab1b, and Rab11a	Genetic instability, MET, and oncogenic transformation	[66]
Evading growth suppressors	circRNA_100284	Inhibition of EZH2/cyclin-D1 gene silencing	[55]
	hsa_circ_0000069	Enhance the expression of the STIL gene	[56]
Avoiding immune destruction	exosomal regulating proteins and miRNAs PD-1, MET, RAF1, BCL2, and mTOR	PD-1 overexpression	[67]
Enabling replicative immortality	hTERT transcript	Translated into a full-fledged enzyme and initiates telomere elongation	[68]
Tumor-promoting inflammation	Not defined	Increasing transcription of genes for inflammation-supporting cytokines and chemokines (IL-6, IL-8 IL-1, and CXCL-8)	[69]
Activating invasion and metastasis	hsa_circ_0000069	Enhance the expression of the STIL gene	[56]
	ANXA1	Not defined	[62]
Inducing or accessing vasculature	TIE2	High expression of VEGF, PDGF-bb, IL-10, IL-6, IL-1β, and TNFα	[70]
	miRNA-21	Activation of PDK1/AKT signaling. Secretion of VEGF, MMP2, MMP9, bFGF, and TGF-β	[53]
Genome instability and mutation	Not defined	Exosomes induce random genetic change	[37,64]
Resisting cell death	miR-224-5p	Inhibition of the oncosuppressor—CMTM4	[61]
Deregulating cellular metabolism	miR-105	Activates the MYC pathway, enhances glycolysis, glutamine decomposition, and detoxifies the metabolites (lactate and NH4+)	[71,72]
Unlocking phenotypic plasticity	ΔNp73	Induction of proliferation potential and chemoresistance	[73]
Nonmutational epigenetic reprogramming	SND1-IT1	Competitively absorb miR-1245b-5p Recruit DDX54 to upregulate USP3 expression SNAIL1 deubiquitination	[58]
Polymorphic microbiomes	Not defined	Not defined	_
Senescent cells	Not defined	Not defined	_

**Table 3 cells-13-00682-t003:** Participation of some EV cargo components in the transformation into cancer-associated support cells with certain phenotypes.

Cells of TME	Tumor Cells EV Factors	Consequence	Reference
Cancer-associated fibroblasts (CAFs)	Piwil2-iCSC	Increase in the expressions of MMP2 and MMP9	[114]
	Gm26809	Not defined	[115]
	miR-146a	Downregulation of the TXNIP gene, activation of the Wnt signaling pathway	[116]
	miR-27a	Directly target the oncosuppressor CSRP2, induction of the ERK and PAK/LIMK/cortactin signaling cascades	[117,118]
	hTERT	Enhancement of telomerase activity	[119]
		Upregulations of αSMA and vimentin	[68]
Tumor-associated macrophages (TAMs)	miR-103a	Decrease in PTEN levels, increased activation of AKT and STAT3	[120]
	miR-29a-3p	Increase in the phosphorylation of STAT1	[121]
	miR-21	Increase in expressions of IL-6 and TNF-α	[122]
Tumor-associated neutrophils (TANs)	KRAS	Upregulation of IL-8 production	[123]
	HMGB1	Activation of the NF-κB pathway	[124]
	circ-CTNNB1	Increase in PD-L1 expression	[125]
Tumor-infiltrated/-associated natural killer cells (TINKs/TANKs)	TGF-β1	Downregulations of activating receptors: NKG2D, NKP30, NKP44, NPK46, and NKG2C	[126]
	NKG2DL	Downregulation of NKG2D expression	[127]
	miR-378a-3p	Decrease in granzyme-B (GZMB) secretion	[128]
Tumor endothelial cells (TECs)	Not defined	Inhibition of TRPV4	[129]
Cancer-associated adipocytes (CAAs)	miR-1304	Regulation of GATA2 gene expression	[130]

**Table 4 cells-13-00682-t004:** Exosome-based TME reprogramming therapy.

Target Cell	Cargo(es)	Outcomes	Cancer Type	Reference
Tumor cells	mir-302s	Reprogramming tumor cells into induced pluripotent stem cells with decreased tumorigenicity	Skin cancer	[226]
	Exosomes derived from human embryonic stem cells	Reprogramming tumor cells into induced pluripotent stem cells with decreased tumorigenicity	Mammary carcinoma, colorectal adenocarcinoma	[227]
CSC	Cell-derived exosomes with osteoinductive potential (OD-EXOs)	Reprogramming cancer stem cells into nontumorigenic cells Enhanced expression of osteogenic-related genes (alkaline phosphatase [ALPL], osteocalcin [BGLAP], and runt-related transcription factor 2 [RUNX2])	Osteosarcoma	[228]
TAM	Antisense oligonucleotide (ASO) targeting STAT6 (exoASO-STAT6)	Reprogramming TAMs toward pro-inflammatory M1 and generation of a CD8 T-cell-mediated adaptive immune response Selectively silences STAT6 expression in TAMs; induces nitric oxide synthase 2 (NOS2)	Colorectal cancer, hepatocellular carcinoma	[229]
	Exosomes derived from M1-type macrophages (M1-Exo)	Reprogramming of TAMs toward pro-inflammatory M1; increased phagocytic function and robust cross-presentation ability	Breast cancer, colon adenocarcinoma	[230]
	Exosomes derived from bone marrow mesenchymal stem cell (BM-MSC), electroporation-loaded galectin-9 siRNA, and surficially modified oxaliplatin (OXA)	Reprogramming TAMs toward pro-inflammatory M1; cytotoxic T lymphocyte recruitment and Treg downregulation	Pancreatic ductal adenocarcinoma (PDAC)	[231]
CAF	Calcipotriol (ligand of vitamin D receptor)	Conversion of activated to quiescent pancreatic stellate cells (myofibroblast-like cells) Increased intratumoral gemcitabine delivery	Pancreatic tumors	[232]
	Dasatinib (PDGFR inhibitor)	Conversion of activated to quiescent fibroblasts	Lung adenocarcinomas	[159]

**Table 5 cells-13-00682-t005:** EVs-based tumor therapy drugs.

Tumor Type	Title of Drug	Phase of Clinical Trials	Short Described	Result	NCT Number
Pancreatic cancer	iExosomes	1	Mesenchymal stromal cell-derived exosomes with KrasG12D siRNA	-	NCT03608631
Hepatocellular carcinoma	exoASO-STAT6 (CDK-004)	1	Delivery of the STAT6 antisense oligonucleotide (ASO) to the myeloid to repolarize macrophages from immune-suppressive M2 to the proinflammatory M1 phenotype	-	NCT05375604
Lung cancer	CSET 1437	2	Immunotherapy involving metronomic cyclophosphamide (mCTX) followed by vaccinations with tumor antigen-loaded dendritic-cell-derived exosomes (Dexs). mCTX inhibits Treg functions restoring T and NK cell effector functions and Dexs are able to activate the innate and adaptive immunity	-	NCT01159288
Head and neck cancer	-	1	Grape exosomes to reduce the incidence of oral mucositis during radiation and chemotherapy treatments	-	NCT01668849
Acute myeloid leukemia	UCMSC-Exo	1	Umbilical-cord-derived mesenchymal stem cells exosomes (UCMSC-Exos) for effectively promoting recovery of myelosuppression	-	NCT06245746
Bladder cancer	-	Early Phase 1	Chimeric exosome vaccine based on dendritic cells or macrophages secretion	-	NCT05559177

## Data Availability

The data that support the findings are available from the corresponding author.

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
