# Peer review of "Tumor Microenvironment Modulation by Cancer-Derived Extracellular Vesicles"

_cells, 2024, doi:10.3390/cells13080682_

Round 1

Reviewer 1 Report

Comments and Suggestions for Authors

The authors focuse on the functions of EVs in the modulation of the TME with a view to how exosomes contribute to the transformation of normal cells, as well as their importance for cancer diagnosis and therapy. Generally, this work will attract broader readership. However, before the work publish, several points need to be addressed.

a) Take-to-home messages and some opening questions should be included in the end of this review.

b) The quality of some figures should be improved.

c) The format of reference should be unified and include more latest literatures.

d) Imagine TME as an ecosystem, the function of EVs in TME is highly recommended to be discussed in the manuscript. 

Author Response

We would like to thank the reviewer for the provided comments. We have revised the manuscript accordingly. Please find the response to the comments below.

COMMENT 1: Take-to-home messages and some opening questions should be included in the end of this review.

ANSWER 1: We have modified the conclusion section of the manuscript. Additionally, we have expanded other sections of the review adding new figures and tables to support the provided information.

COMMENT 2: The quality of some figures should be improved.

ANSWER 2: We have improved the quality of the figures and additionally added graphical abstract as well as new figures and tables to the manuscript.

COMMENT 3: The format of reference should be unified and include more latest literatures.

ANSWER 3: We have corrected the reference list and added more latest publications.

COMMENT 4: Imagine TME as an ecosystem, the function of EVs in TME is highly recommended to be discussed in the manuscript. 

ANSWER 4: We have added description of tumor microenvironment as an ecosystem and further implications of it to the conclusion section.

Reviewer 2 Report

Comments and Suggestions for Authors

Dear Authors,

congratulations for your interesting and well described Review. Current knowledge on the modulation of tumor microenvironment by cancer-derived EVs is well explained and it is supported by a robust and updated bibliography.

But you have to check the references very well. In the section of Reference List you have repeated twice two references, (please see n. 50, 51, 81 and 82 and correct). In the text some numbers of the References in the brackets are bold others not, please standardize the style.

Author Response

We would like to thank the reviewer for the provided comments. We have revised the manuscript accordingly. Please find the response to the comments below.

COMMENT 1: But you have to check the references very well. In the section of Reference List you have repeated twice two references, (please see n. 50, 51, 81 and 82 and correct). In the text some numbers of the References in the brackets are bold others not, please standardize the style.

ANSWER 1: We have revised the references adding more latest publications to it. As well we have corrected and standardized the references in the manuscript.

Reviewer 3 Report

Comments and Suggestions for Authors

This manuscript extensively reviews the impact of cancer-derived extracellular vesicles (EVs), particularly exosomes, on the tumor microenvironment (TME). Emphasizing the vital role of TME in tumorigenesis and therapy resistance, the study explores how EVs transfer various molecules to reshape normal cells within the TME into pro-tumorigenic entities. It discusses EV classification, contents, and their role in intercellular communication. The review underscores the complexity of interactions within the TME, highlighting the transformation of normal cells like cancer-associated fibroblasts and tumor-associated macrophages. The conclusion advocates for a deeper understanding of these interactions, suggesting that targeting the TME in therapeutic strategies could enhance treatment efficacy by disrupting tumor resistance mechanisms.

Minor revisions

Please standardize references throughout the text according to a chosen format; inconsistencies in formatting, such as bold and italic styles, are observed.

Consider changing Section 5 to "Conclusion" for better clarity in the manuscript's structure.

Expand the Conclusion section to enhance its depth and organization. Given the absence of a separate Discussion section, the Conclusion should not only summarize but also provide a thoughtful evaluation of the presented information, emphasizing the significance and relevance of the research. Ensure it guides the reader effectively, making a compelling argument to support the conclusion.

Comments on the Quality of English Language

While this review offers a thorough exploration of the issue and exemplifies an extensive literature analysis, it would benefit from a proofread for English grammar and syntax. The text occasionally lacks a natural flow, with sentences being too short or overly lengthy. Consider refining the writing style to improve overall coherence and readability.

Author Response

We would like to thank the reviewer for the provided comments. We have revised the manuscript accordingly. Please find the response to the comments below.

COMMENT 1: Please standardize references throughout the text according to a chosen format; inconsistencies in formatting, such as bold and italic styles, are observed.

ANSWER 1: We have corrected the references adding more latest publications. Additionally, we have standardized the references in the manuscript.

COMMENT 2: Consider changing Section 5 to "Conclusion" for better clarity in the manuscript's structure.

ANSWER 2: We have modified the conclusion section.

COMMENT 3: Expand the Conclusion section to enhance its depth and organization. Given the absence of a separate Discussion section, the Conclusion should not only summarize but also provide a thoughtful evaluation of the presented information, emphasizing the significance and relevance of the research. Ensure it guides the reader effectively, making a compelling argument to support the conclusion.

ANSWER 3: In the conclusion section we tried to be more concise. In the revised manuscript we have expanded other sections adding new figures and tables (for example the table demonstrating therapeutic applications of EVs) to emphasize the significance of the research.

COMMENT 4: While this review offers a thorough exploration of the issue and exemplifies an extensive literature analysis, it would benefit from a proofread for English grammar and syntax. The text occasionally lacks a natural flow, with sentences being too short or overly lengthy. Consider refining the writing style to improve overall coherence and readability.

ANSWER 4: We have asked the english speaking colleague to refine the writing style of the manuscript and have done this.

Reviewer 4 Report

Comments and Suggestions for Authors

In the present review, Ten et al. provided a detailed understanding of the role of EVs derived from cancer cells in the modulation of TME. The review is full of useful information; however, a few modifications would make it far better. My comments are appended below:

1. The authors could provide the biogenetic mechanism of different types of EVs in a pictorial form.

2. It would be nice to include a section where the role of cancer-derived EVs on transferring the aggressive cancerous properties to non-aggressive cancer cells is discussed. A few literatures demonstrating these should be cited (PMID: 31341019, PMID: 30129687, PMID: 29743547 etc.). 

3. Section 1-3 could be shown in a tabular view.

4. The role of cancer-derived EVs as biomarkers regarding the modulation of TME could also be discussed briefly.

5. It would be nice to provide the EVs related ongoing clinical trials in cancer in a tabular form. 

Author Response

We would like to thank the reviewer for the provided comments. We have revised the manuscript accordingly. Please find the response to the comments below.

COMMENT 1: The authors could provide the biogenetic mechanism of different types of EVs in a pictorial form.

ANSWER 1: We have added the new figure depicting the biogenetic mechanism of various types of EVs.

COMMENT 2: It would be nice to include a section where the role of cancer-derived EVs on transferring the aggressive cancerous properties to non-aggressive cancer cells is discussed. A few literatures demonstrating these should be cited (PMID: 31341019, PMID: 30129687, PMID: 29743547 etc.). 

ANSWER 2: We have added this information into the revised manuscript and also included the suggested papers into the reference list.

COMMENT 3: Section 1-3 could be shown in a tabular view.

ANSWER 3: To improve the manuscript we have added graphical abstract as well as new figures and tables.

COMMENT 4: The role of cancer-derived EVs as biomarkers regarding the modulation of TME could also be discussed briefly.

ANSWER 4: We have added the reference that the EVs could be employed as biomarkers in cancer diagnostics.

COMMENT 5: It would be nice to provide the EVs related ongoing clinical trials in cancer in a tabular form.

ANSWER 5: We have added a new table that describes the application of EVs in clinical trials.

Reviewer 5 Report

Comments and Suggestions for Authors

This is a comprehensive review paper and it is timely and significant. Tumor microenvironment modulation by cancer-derived extracellular vesicles is an important area of research. The description of topics and organization of five figures and two tables are appropriate.

Comments on the Quality of English Language

Moderate editing of English language is required. The manuscript can be much improved.

Many references cited do not have journal names, e.g. 48, 49, 51, 54, 56, 57, 221-225. This is not acceptable. Please make sure references are cited correctly and abbreviations are spelled out.

Author Response

We would like to thank the reviewer for the provided comments. We have revised the manuscript accordingly. Please find the response to the comments below.

COMMENT 1: Moderate editing of English language is required. The manuscript can be much improved.

ANSWER 1: We have asked the english-speaking colleague to refine the manuscript and this was performed.

COMMENT 2: Many references cited do not have journal names, e.g. 48, 49, 51, 54, 56, 57, 221-225. This is not acceptable. Please make sure references are cited correctly and abbreviations are spelled out.

ANSWER 2: We have corrected the reference list adding latest publications. Additionally, we have standardized the references in the text.

Reviewer 6 Report

Comments and Suggestions for Authors

The article written by Ten and colleagues aimed to provide an overview of the influence of cancer-derived extracellular vesicles (EVs) on cells present in the tumor microenvironment (TME). The authors presented a comprehensive account of the role of EVs, ranging from cancer stem cells (CSC) to therapeutic applications using EVs. The illustrations provided in the article facilitate easy understanding of the concepts discussed. However, some sections, particularly section 3, could be shorter and more focused on the impact of EVs on the TME. Section 4, which deals with therapeutics, is an interesting topic that can be elaborated further. In Line 40, please mention the version of the MISEV update.

Comments on the Quality of English Language

minor editing, for e.g. Line 48, do you mean lipid membrane bilayer? Line 79 to 82, do you mean hypoxic condition increased the released of EV by tumor cells?

Author Response

We would like to thank the reviewer for the provided comments. We have revised the manuscript accordingly. Please find the response to the comments below.

COMMENT 1: However, some sections, particularly section 3, could be shorter and more focused on the impact of EVs on the TME. Section 4, which deals with therapeutics, is an interesting topic that can be elaborated further. In Line 40, please mention the version of the MISEV update.

ANSWER 1:  We have revised the manuscript adding latest publications, new figures as well as new tables (for example the table describing the clinical applications of EVs). Additionally, we added the reference of the latest MISEV version.

COMMENT 2: minor editing, for e.g. Line 48, do you mean lipid membrane bilayer?

ANSWER 2: We have corrected to lipid membrane bilayer.

COMMENT 3: Line 79 to 82, do you mean hypoxic condition increased the released of EV by tumor cells?

ANSWER 3:  We have corrected this sentence as suggested.

Round 2

Reviewer 4 Report

Comments and Suggestions for Authors

The authors adequately addressed all my concerns. I have no further questions.

Author Response

We would like to thank the reviewer for his work and provided valuable comments.

Reviewer 6 Report

Comments and Suggestions for Authors

Line 214, "malignization", do you mean "malignant transformation"?

Line 234-237, "... subpopulations of tumor cells with new phenotypes are formed...". Please clarify what does "new phenotypes" mean? The sentence "Thus, EV-dependent transfer..." has no link to the paragraph. Please clarify.

Comments on the Quality of English Language

Extensive editing of English language required

Author Response

We would like to thank the reviewer for the provided comments. Please find the reply for the comments below.

COMMENT 1: Line 214, "malignization", do you mean "malignant transformation"?

ANSWER 1: This was corrected to "malignant transformation" throughout the manuscript.

COMMENT 2: Line 234-237, "... subpopulations of tumor cells with new phenotypes are formed...". Please clarify what does "new phenotypes" mean? The sentence "Thus, EV-dependent transfer..." has no link to the paragraph. Please clarify.

ANSWER 2: We have corrected this paragraph and reformulated the sentences.